

# The NCAS Mobile Dual-Polarisation Doppler X-Band Weather Radar (NXPol)

Ryan R. Neely III[1,2], Lindsay Bennett[1,2], Alan Blyth[1,2], Chris Collier[1,2], David Dufton[1,2], James Groves[1,2], Daniel Walker[1,2], Chris Walden[1,3,4], John Bradford[1,3,4], Barbara Brooks[1,2], Freya Lumb[1,2], John Nicol[1], Ben Pickering[1,2]

[1]National Centre for Atmospheric Science, University of Leeds, Leeds, UK
[2]School of Earth and Environment, University of Leeds, Leeds, UK
[3]Rutherford Appleton Laboratory, Didcot, UK
[4]Chilbolton Facility for Atmospheric and Radio Research (CFARR), Chilbolton, Hampshire, UK

*Correspondence to*: Ryan R. Neely III (r.neely@ncas.ac.uk)

**Abstract.** In recent years, mobile dual-polarisation Doppler X-band radars have become a prevalent part of the atmospheric scientist's toolkit for examining cloud dynamics and microphysics and making quantitative precipitation estimates. Here we describe the National Centre for Atmospheric Science (NCAS) mobile X-band Dual-polarisation Doppler weather radar (NXPol) and the infrastructure used to deploy the radar and provide an overview of the technical specifications. It is the first radar of its kind in the United Kingdom. The NXPol is a Meteor 50DX manufactured by Selex-Gematronik (Selex ES GmbH), modified to operate with a larger 2.4 m diameter antenna that produces a 0.98° half-power beam width and without a radome. We provide an overview of the technical specifications of the NXPol with emphasis given to the description of the aspects of the infrastructure developed to deploy the radar as an autonomous observing facility in remote locations. To demonstrate the radar's capabilities, we also present examples of its use in three recent field campaigns and its ongoing observations at the Chilbolton Facility for Atmospheric and Radio Research (CFARR).



## 1 Introduction

Polarimetric radars are powerful tools for meteorological studies. The diverse quantities observed by polarimetric radars can provide significant insights into the evolution of clouds and precipitation (e.g. Fabry, 2015). Thus, small and or mobile dual-polarisation Doppler X-band radars have become ubiquitous tools for examining cloud microphysics and dynamics as well as making quantitative precipitation estimates (QPE) (Wurman et al. 1997; Matrosov et al. 2005; Wang and Chandrasekar, 2010). Currently, a significant number of such radars exist in the operational and research sectors to address a broad range of scientific goals pertaining to atmospheric physics and hydrometeorology (Maki et al. 2005; Bluestein et al. 2007; 2014; Kato, A. and Maki, 2009; Pazamny et al. 2013; Forget et al. 2016; Mishra et al. 2016; Antonini et al. 2017). Use of such radars notably includes recent field campaigns such as PECAN (Plains Elevated Convection At Night, Geerts et al., 2016), where a variety of mobile radars (both X-band and C-band) from multiple institutions were used collaboratively to achieve complex goals successfully. In the United States, where mobile research radars are more numerous, large multi-institution observational campaigns, similar to PECAN, occur several times a decade (e.g. the second Verification of the Origins of Rotation in Tornadoes Experiment (VORTEX-2), Wurman et al. (2012)). Mobile radars are also used as a teaching resource, for example, the University of Oklahoma SMART (Shared Mobile Atmospheric Research and Teaching) radar (Biggerstaff et al., 2005). Thus, it is difficult to understate the role of such instrumentation in hydrometeorology and atmospheric research.

Here we describe the NCAS Mobile X-band dual-polarisation Doppler weather radar (NXPol) shown in Figure 1 and the supporting infrastructure structure that has been developed to support the radar when on deployment. The NXPol is the first dual-polarisation mobile radar in the United Kingdom. The supporting infrastructure has been developed to create a robust facility that may be operated remotely with minimal staff. As such, the NXPol has developed into a semi-operational observing system facility that has the significant capabilities present in both traditional research radars used for intensive operational periods (IOPs) and radars operated as part of national networks. In addition to the technical description, examples of NXPol in 3 differing campaigns are shown, as well as an example of its ongoing use at the Chilbolton Facility for Atmospheric Radar Research (CFARR). The NXPol is part of the pool of mobile instruments that make up the UK NCAS Atmospheric Measurement Facility (NCAS-AMF, https://www.ncas.ac.uk/index.php/en/about-amf) so it is available for use by the community according to the procedures set out by NCAS-AMF.




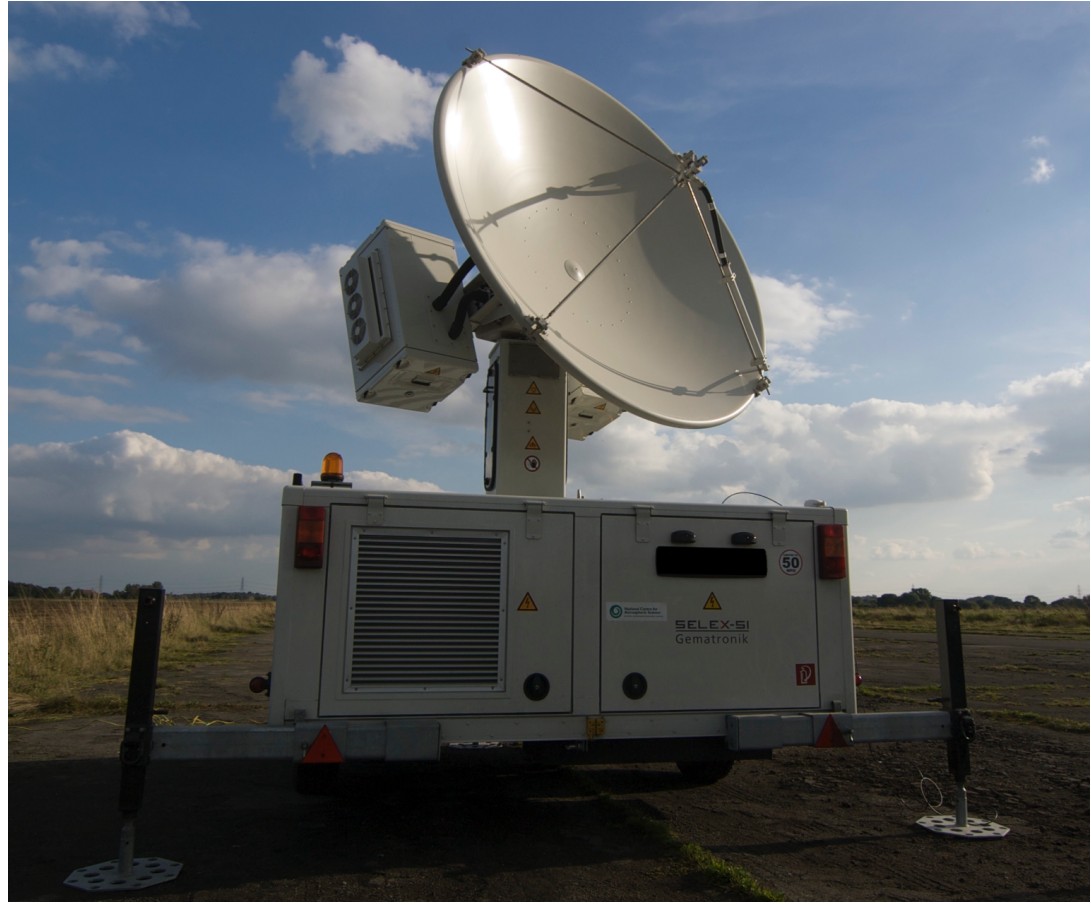

**Figure 1: Photograph of the NXPol collecting data at Burn Airfield near Selby, UK. Here the NXPol is deployed using only its**
**trailer as a platform.**





**2 Technical Summary of the NXPol**
The NXPol is a modified mobile Meteor 50DX (Selex ES GmbH) X-band, dual-polarisation, Doppler weather radar. The
radar is a magnetron based system and operates at a nominal frequency of 9.375 GHz (~3.2 cm). A detailed description of
the development of this class of Selex radars is given by Borgmann et al. (2007). The radar is capable of measuring areal
precipitation, radial winds and properties of cloud and precipitation particles. It can also detect clear-air echoes, including
biota, at close range by scanning at slower speeds and optimising the transmitter and receiver. Similar radars (including the
newer Meteor 60DX) are utilised by national weather services and research centres throughout the world. Table 1 provides a
summary of the technical characteristics of the NXPol.

Like all standard mobile Meteor 50/60DX radars, NXPol is transportable. The radar is constructed on a wheeled platform
that is approved for towing on roads in the European Union by a 4x4 vehicle and can also be lifted by a crane. This trailer
includes a generator to provide necessary power and the communications infrastructure to operate and monitor the radar
remotely for up to 24 hours. This mobility makes NXPol a highly versatile tool for studying a diverse array of atmospheric
phenomenon across the globe. The main difference between NXPol and the standard mobile Meteor 50/60DX is that the
NXPol has been fitted with a larger 2.4 m diameter antenna that produces a 0.98° half-power beam width. The NXPol is
operated without a radome, which is beneficial for eliminating radome attenuation effects, but extra care is required during
transport, and long-distance shipping may need the antenna and external waveguides to be removed. The decision to fit
NXPol with a larger antenna was made to support the ability to make higher resolution observations of convective clouds.
In comparison, the standard mobile Meteor 50/60DX has a 1.8m antenna that produces a 1.3° half-power beam width and is
usually operated with a radome. In addition to its increased spatial resolution, NXPol is also advantageous for use in the
observation of cloud evolution because of its rapid scanning capabilities; up to 36 degs$^{-1}$.




**Table 1. Technical characteristics of the NXPol.**

| Parameter | Specifications |
|---|---|
| Frequency | 9.375 GHz |
| Transmitter Type | Coaxial Magnetron |
| Peak Transmit Power | ~80 kW |
| Average Power | ~80W |
| Dual-Polarisation Mode | Simultaneous H & V |
| Digital Receiver and Signal Processor | GDRX®4 |
| Receiver Linearity | 90 dB +/- 0.5 dB |
| Antenna Diameter | 2.4m |
| Half Power Antenna Beam Width | 0.98° |
| Antenna Gain | 44dB |
| Radome | None |
| Elevation Scan Range | -1 to 181° |
| Azimuthal Scan Range | 0 to 360° |
| Position Accuracy | ±0.1° |

**2.1 Operations**
The NXPol can be operated via a remote computer (e.g. a laptop or server) that connects by wireless, ethernet or 3G to PCs
onboard the NXPol's trailer unit. The operational software allows the user to set up the radar for deployment and schedule
the scanning sequence. Ravis® is the maintenance and calibration software used for system diagnostics and testing, as well
as real-time data visualisation. Ravis® includes an automatic sun tracking tool for alignment of the system. Rainbow®5 is
the scan scheduling, data visualisation and analysis software, providing near real-time product and image generation. As
shown in Table 2, the NXPol is highly configurable with regards to the pulse width, PRF and scan pattern and can be tailored
to address the specific scientific question being examined. Bold values in Table 2 indicate the typical parameter settings used
in the examples shown in Section 3. Signal retrieval, analysis and data storage are performed by the GDRX®4 digital
receiver and signal processor.





**Table 2. Parameter settings. Boldface indicates settings typically used for operations.**

| Parameter | Specifications |
|---|---|
| Pulse Width | 0.5μs, 1μs, 2μs (**1μs**) |
| Pulse Repetition Frequency (PRF; Single or Dual Modes) | 250-2000 Hz (**1000 Hz single-PRF mode, 1000Hz/800Hz dual-PRF mode**) |
| Dual PRF Mode | 3/2, 4/3, 5/4 (**5/4**) |
| Unambiguous Velocity using single-PRF | ±8m/s - ±16 m/s (**±8m/s**) |
| Unambiguous Velocity using dual-PRF | ±8m/s - ±64 m/s (**±32m/s**) |
| Range Resolution | 50m-300m (**150m**) |
| Maximum Range Gates | 2000 (**2000**) |
| Maximum Operating Range | 600 km (**150 km**) |
| Antenna Speeds | 0 to 36° s$^{-1}$ (**~13-24° s$^{-1}$**) |


Note that the NXPol operates only using the hybrid polarisation basis, also known as the simultaneous transmit and receive
(STAR) mode (i.e. it splits the transmitted signal into two parts and simultaneously transmits and receives horizontal (H) and
vertical (V) polarisations) (Chandrasekar and Bharadwaj, 2009). This mode operates under the assumption that the cross–
polarisation signals are weak in comparison to the co-polar signals and are therefore negligible (Wang and Chandrasekar,
2006)). As the cross-polar signals are not measured, observations of the linear depolarisation ratio (LDR) are not available.
The benefit of STAR mode is that the NXPol has a much simpler and robust hardware design because it avoids switching
between H and V polarisation on a pulse-to-pulse basis (Doviak et al. 2000; Bringi and Chandrasekar 2001). STAR mode
operations also lead to less noisy measurements of differential reflectivity ($Z_{DR}$) and other quantities while operating at rapid
scan speeds.
The dual-polarisation capability of the NXPol allows for the retrieval of many additional geophysically-related variables.
This additional information helps to provide insight into the size and shape of precipitation, enhanced target identification as
well as the assessment of attenuation and propagation effects (Bringi and Chandrasekar, 2001; Kumjian 2013a;b;c; Fabry,
2015). The NXPol's polarimetric ability also enables many alternative methods for quantitative precipitation estimation,
which are demonstrated in Section 3 (Deiderich et al. 2015; 2015). One unique aspect of the NXPol is the implementation of
the retrieval of the degree of polarisation (DOP). DOP is a relatively unexplored variable with respect to atmospheric
phenomenon, but previous examinations have shown that it has similar properties as the co-polar correlation coefficient





when classifying hydrometeors (Galletti et al. 2007; 2012). Galletti et al. (2012) note that DOP is advantageous compared to the co-polar correlation coefficient for STAR mode radars like the NXPol because it retains its physical meaning even when observing scatterers that are cross polarising (i.e. with linear depolarisations ratios that are greater than zero).

During operations, scan strategies are tailored to the application but typically sample a volume out to 150 km in range every 5 minutes. The typical volume includes ~10 PPI scans between 0.5° and 30° of elevation and a calibration scan at 90°. All data are recorded as moments in Selex's Rainbow®5 format (a flavour of XML). This format is easily utilised by common open-source analysis software packages (Heistermann et al. 2015) such as the LIDAR RADAR Open Software Environment (LROSE) that is provided by the Earth Observing Laboratory within the U.S.'s National Center for Atmospheric Research (NCAR) (Dixon et al. 2012; 2013), the Python Atmospheric Measurement Radiation (ARM) Climate Research Facility Radar Toolkit (PyART) (Helmus and Collis, 2016) and the Open Source Library for Weather Radar Data Processing (wradlib) (Heistermann et al. 2013). The NXPol has the capability of collecting raw IQ data for post-processing, but this is typically not done due to the size of the dataset.

Once a volume is collected, the raw data are backed up locally and transferred to a central NCAS data storage facility if internet capacity allows as described in Section 2.2.  In addition to storing the data for later analysis, Rainbow®5 generates several quick-look images in real-time (tailored to the application of the radar). The quick-look images are transferred to a central server where they are uploaded onto a web catalogue to disseminate the observations in near real-time and enable easy examination of past observations. Figure 2 depicts an example of a real-time image and corresponding catalogue page. Such near-real-time quick-look charts were crucial in the two field campaigns discussed later for changing scan patterns and directing aircraft. They were also invaluable to the NXPol's operators as well as the forecasters at the Scottish Environmental Protection Agency and the U.K.'s Met Office during the six-month-long Radar Applications in Northern Scotland (RAINS) campaign in 2016 (Section 3.3).

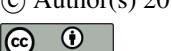



**Figure 2. Example of the data catalogue used to monitor the NXPol observations in near-real time during the RAINS project**
**described in Section 3.3. The background shows a collection of a set of images from a single day while the foreground highlights an**
**example of a near-real-time image produced by Rainbow®5.**

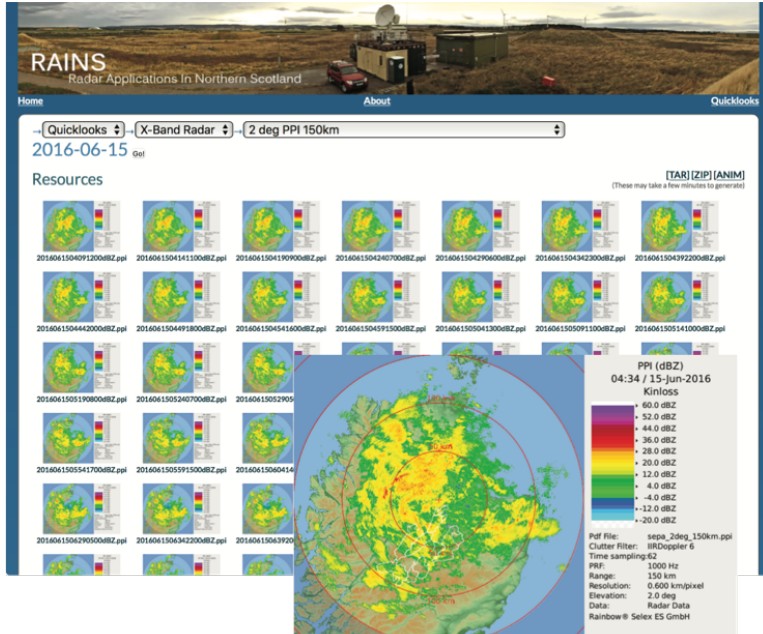






### 2.2 Deployment Setup

The operational requirements dictated by the strategic and project-specific scientific goals of the NXPol have led to the development of bespoke infrastructure to support the radar during operations. The primary requirements for deploying the NXPol radar are visibility, security, power and internet access. Considering these options, the NXPol may be deployed using solely its integrated trailer (as in Figure 1) or in conjunction with a platform structure as depicted in Figure 3. The platform setup is based on a similar scheme employed by Selex ES GmbH for the NXPol's deployment during the Single European Sky ATM (Air Traffic Management) Research (SESAR) campaign in 2015 at Braunschweig Airport near Hanover, Germany. The setup has the major advantage of lifting the radar off the ground to provide greater visibility. It also makes security less problematic.

The platform consists of a 20-foot standard shipping container and a 20-foot office container set side by side along their long axis. To provide the necessary structural strength to support the weight of the NXPol, on top of each of the containers is a 20-foot platform container (also known as a 'flat rack'). Using standard shipping containers and platforms dramatically reduces engineering time and cost during deployments. Also, because of their global ubiquitousness, the elements needed to construct a similar platform can be sourced locally. This further reduces deployment costs. To provide safe access to the radar while it is on the platform, a staircase and railing are constructed from standard scaffolding materials as shown in Figure 3. Also attached to the platform structure are the various pieces of hardware that support a long-term autonomous deployment of the NXPol; lightning protection, a satellite internet connection, security camera and local weather station.

In addition to providing a platform for the NXPol, the office unit provides space for the additional IT infrastructure needed for NXPol's autonomous operation (described below). The office also provides a base of operations for staff while on site during remote fieldwork. The office is particularly useful during observational campaigns that involve the coordinated operation of the radar and an aircraft (such as the ICE-D campaign described in Section 3.2). During such campaigns, staff can monitor and direct the radar's observations in real-time and communicate with the aircraft to help target the observations.





**Figure 3. NXPol deployed at Chilbolton Observatory, Hampshire, UK. Seen in the picture is the 20' shipping container, 20' office**
**container, two 20' platform containers and scaffolding used to construct a platform for the radar.**

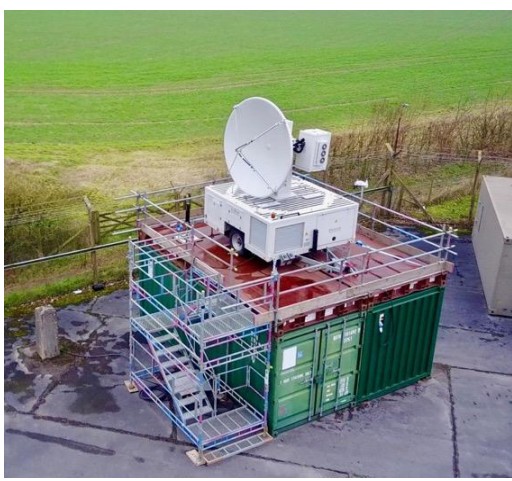


There is also the need in the scientific community for the collection of statistically meaningful observations over a wide
range of synoptic conditions. This requirement has necessitated the move to semi-permanent, continuous and autonomous
operations that last for many months. The RAINS project had such a need and now the radar has been based at CFARR for
several months and will remain there while not being used for field-campaign work.

Table 3 summarises the operational requirements of the NXPol. Data and power availability vary depending on the
deployment. Typically, when the NXPol is deployed for less than 24 hours, the onboard generator supplied by an 80 L fuel
tank provides all electricity. An onboard 3G mobile data connection or satellite link provide internet connectivity. When the
NXPol is deployed using the container platform, mains electricity is connected to the radar's electrical grid and the onboard
generator acts as a backup power supply that is automatically started upon loss of mains power. Additionally, the 3G mobile
data connection is supplemented with a local area network connection or a satellite internet connection. This allows for
more robust autonomous and remote operation of the system.




**Table 3. Operational conditions and logistical requirements of the NXPol.**

| Conditions | Specifications |
|---|---|
| Max Operational Wind Speed without Radome | 56 mph (90km/h) |
| Electrical Supply | 3-phase 32A Service or Onboard Diesel Generator |
| Power Consumption | 8kW (average), 12kW (max) |
| Operating Temperatures | -10C to 35C |
| Total Weight, Nose weight | 2800kg, 120kg |
| Width (without supports, with supports) | 2.550m, 3.560m |
| Height (0 degs, 90 degs) | 3.995m, 4.250m |
| Max drive Speed | 50mph (80km/h) |


The IT infrastructure needed for the NXPol's autonomous operation includes a server that provides a gateway for
communicating with the radar and data backup. Figure 4 summarises the IT strategy.  In addition to communications, the
infrastructure includes a local weather station to primarily monitor wind speeds, a video camera to monitor the radar's
movement and an Uninterruptible Power Supply (UPS) for the server. Data production is on the order of between 5 and 9 Gb
per day. During long-term remote operations the data can be backed-up using a commercial satellite internet system if
available, although it may be cost-prohibitive if there is not an unmetered period (typically in the early hours of the morning
local time). The onboard 3G connection provides redundancy and/or remote control, if local signal strength permits, but it is
not practical to backup bulk data via this route.  If near-real-time remote raw data access is required, a suitable Internet
connection is necessary. Quick-look charts as shown in Figure 2 use considerably less data and are therefore logistically
simpler, potentially allowing selective download of raw data over a lower-bandwidth connection.  Data are backed-up locally
to a Network Attached Storage (NAS) system in the office container in medium- to long-term deployments.



**Figure 4. Schematic of the IT infrastructure used by the NXPol when on deployment.**

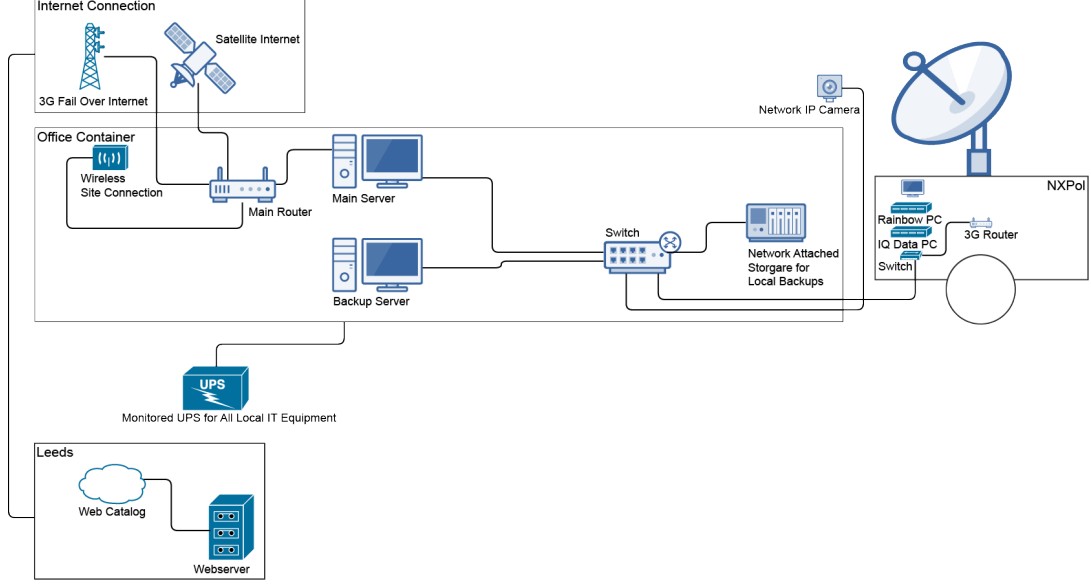




## 2.3 Safety

An important consideration when deploying the NXPol radar is the protection of both operators and the public from exposure to transmissions. The location of the deployment site is determined in conjunction with the required safety distance specified by a radiation exposure assessment. The International Commission on Non-Ionizing Radiation Protection (INCIRP) specifies that the maximum continuous exposure to radiation at frequencies between 2 GHz and 300 GHz should not exceed $10 Wm^{-2}$ in areas where the general population has access and $50 Wm^{-2}$ for occupational exposure, averaged over a period of 6 minutes (Ahlbom et al. 1998). When deployed on the ground, a safety barrier must be constructed or measures put in place to prevent access within the distance which the exposure threshold would be exceeded as determined by the radiation assessment (e.g. if the radar dish stops scanning). When NXPol is situated on a platform and is scanning and operating as scheduled, there is no risk to people on the ground at any distance from the radar and hence is another benefit of this method of deployment. If the radar is unmanned, a contingency plan must be considered in the event the system malfunctions and stops scanning but continues transmitting.

The second major safety consideration is the operation of the system in high winds. Without a radome the maximum operational wind speed is 56mph. The weather station continuously monitors the wind speed and notifies operators via text and email alerts when a set threshold (typically below the 56mph maximum limit to allow for gusts) is exceeded. Operators closely monitor the conditions during forecasted events and, in the case of significant winds, interrupt the scan schedule to move the antenna into the vertical position (which provides the least wind resistance) and activate the locking stow pin to prevent movement. In addition to winds, NXPol's temperature must be monitored carefully to avoid operations below -10C and above 35C as there is no radome to provide a conditioned environment for the transmitter and receiver equipment boxes located behind the antenna. This operational range also limits the regions where the NXPol may be deployed.



## 3 Example Deployments and Observations

Below, four examples of the use of NXPol are given. Descriptions are provided to highlight the utilisation of the radar to achieve the scientific aims of each project.

### 3.1 COPE

NXPol was utilised for the first time in the COnvective Precipitation Experiment (COPE) held in the vicinity of Davidstow, Cornwall during July and August, 2013. Three aircraft, including the Facility for Airborne Atmospheric Measurement (FAAM) BAe-146 aircraft, and other ground-based instruments were also deployed; see Leon at al., (2015). The principal aim of the project was to understand the physical processes involved in the production of heavy convective precipitation that could result in flash flooding. Ultimately, predictions of heavy precipitation and potential flash floods by Numerical Weather Prediction (NWP) models will be improved as a result of the new knowledge and understanding of physical processes. Several flash flooding events have previously occurred in the region, the most notable in recent years being the Boscastle flood of 2004 (Golding et al. 2005).   The role of the radar was to determine (a) the altitude of the first echoes; (b) the rate of development of the reflectivity echoes; (c) the spatial and temporal distribution of the main echoes; (d) the particle types from dual-polarisation parameters (e.g. warm rain or graupel), and (e) the maximum intensity of the precipitation.

NXPol collected data during 16 IOPs covering a variety of synoptic and microphysical conditions including heavy precipitation from shallow clouds (warm rain only) and several cases of deep convection along semi-organised convective lines with similarities to the Boscastle event. An example of the convective clouds that formed along a convergence line (at 20 km range between S and SE) and observed elsewhere on 3 August, 2013 is shown in Figure 5.  Note that Figure 5 and all following figures were created using software developed in NCAS that is based on the Py-ART software suite (Helmus and Colis, 2016).  The rainfall rates (Figure 5d) were derived from the unfiltered and uncorrected calibrated horizontal reflectivity ($Z_H$, Figure 5a) by first applying a second trip filter and a fuzzy logic clutter filter as described by Dufton and Collier (2015). In addition to these corrections, a correction for partial beam blocking and attenuation ($A_H$) have also been applied. From this corrected $Z_H$, rainfall rate was retrieved using the Marshall-Palmer relation ($R(Z)=aZ^b$, with a=200 and b=1.6 as is used by the UK Met Office) to derive rain rate for their operational network of C-band radars (Marshall and Palmer, 1948). For access to the observations made with the NXPol during COPE, please see the Centre for Environmental

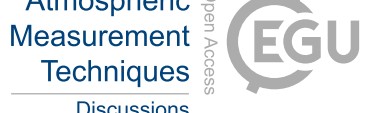



Data Analysis (CEDA) archive for the campaign at http://data.ceda.ac.uk/badc/microscope/data/ncas-mobile-xband-
radar/version-2/ (Blyth et al., 2013).

**Figure 5. Example of observations made by NXPol (located at the centre black dot) at 0.5° elevation on 3 August 2013 at 1332**
**UTC showing: a) calibrated but unfiltered and uncorrected horizontal reflectivity (Shown as to display the importance and impact**
**of the data processing) , b) calibrated, filtered and corrected differential reflectivity, c) specific horizontal attenuation ($A_H$) and d)**
**rainfall rates derived using the Marshall-Palmer relation (R(Z)=a$Z^b$, with a=200 and b=1.6.  The sub-panel in d) shows an**
**expanded section of the line of intense rainfall (>120mm/hr in some pixels; please note the expanded colorbar to the top of the sub-**
**panel) to the southeast of the radar.  Range rings are drawn every 10km.**

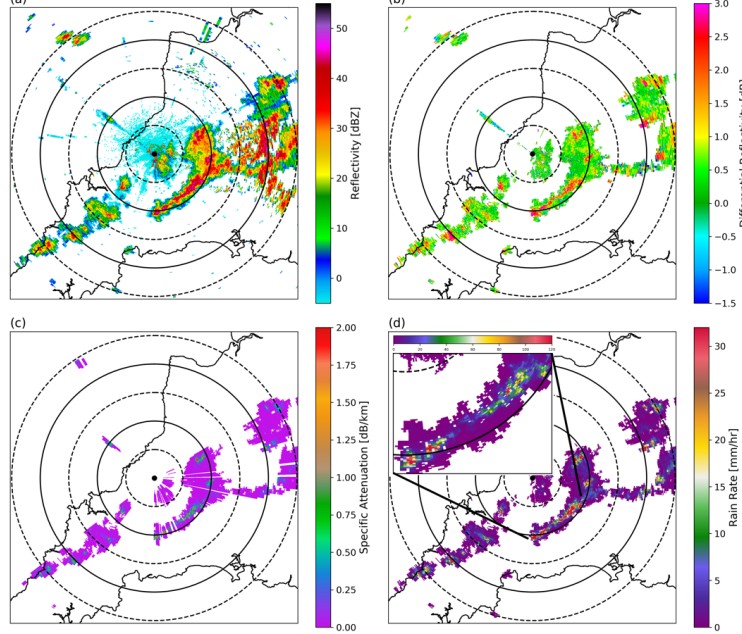






**3.2 ICE-D**
NXPol was deployed at Praia, Cape Verde (14°55′N 23°31′W) during July and August 2015 in the UK's Ice in Clouds
Experiment-Dust (ICE-D). The goal of ICE-D was to determine how desert dust affects primary nucleation of ice particles in
convective and layer clouds and the subsequent development of precipitation and glaciation of the clouds. In addition to
NXPol, the FAAM BAe-146 research aircraft and the University of Manchester ground-based aerosol laboratory were
deployed.

The main objective of NXPol was to provide the spatial and temporal distribution of the clouds, to identify suitable cloud
regions for the aircraft to sample and to provide coordinated observations of the development of precipitation within about
100 km of the island. Two modes of data collection were implemented dependent on the synoptic conditions and location of
cloud development. In "surveillance mode", NXPol was configured to maximise its observable range. In this mode,
observations were made out to 300 km at several low elevations. An example surveillance mode PPI observed on 23 August
2015 using the surveillance mode is given in Figure 6. Use of the radar in this mode was found to be invaluable for near-term
mission planning and directing the use of the FAAM once it was airborne.   For suitable clouds at closer range, NXPol
operated in "data-collection mode", providing higher spatial and temporal resolution observations; volumes of 12 elevations
from 0.5 up to 12 degrees were collected out to a range of 150 km similar to COPE.
**Figure 6. Example of a surveillance mode PPI observed by NXPol at 2119UTC on 23 August 2015 while on the ICE-D deployment**
**in Praia, Cape Verde. Range rings are drawn every 50km.**

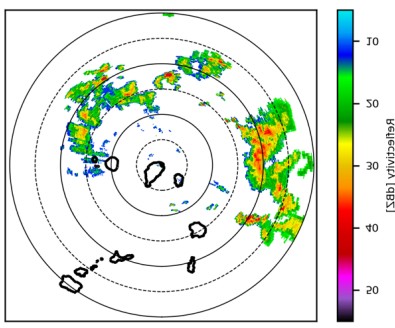




### 3.3 Radar Applications in Northern Scotland (RAINS)

COPE and ICE-D are examples of the use of the NXPol for traditional IOP based operations. This section highlights the use of the NXPol for semi-permanent operations. Previous studies have shown the value of operating a mobile polarimetric X-band radar in coastal regions to fill gaps in the coverage of national operational radar networks (Matrosov et al. 2005). Matrosov et al. (2005) found that the NOAA X-band radar (9.34 Hz, 30kW peak power) was effective in covering an area up to 40–50 km in radius offshore adjacent to a region that is prone to flooding during wintertime land landfalling Pacific storms.  More recently, the Collaborative Adaptive Sensing of the Atmosphere (CASA) Engineering Research Center's X-band dual-polarisation radar network has shown the utility of short-range radars at making high-resolution observations of rainfall that are close to the ground over a variety of conditions (Wang and Chandrasekar, 2010).

During the RAINS campaign, the NXPol was installed at Army Base 39 Engineer Regiment, Kinloss, northeast Scotland from January 2016 to August 2016. This deployment was a joint project between NCAS, the Scottish Environment Protection Agency (SEPA), the University of Leeds and the UK Met Office with the goal of examining the value of additional and higher-resolution radar observations in this region for creating more accurate QPE and flood forecasts. Beyond just improving radar coverage in Northern Scotland, the data collected from the NXPol is also being used to examine the specific improvements in QPE that dual-polarisation observations can provide hydrological models in this region, which is characterised by low melting levels (i.e. low bright bands) and mountainous terrain. In Figure 7 we show an example of two differing QPEs during a typical precipitation event during the deployment. The observations and the two rainfall rate retrievals are shown here to highlight the potential differences in rainfall rate methods that are being explored as part of RAINS.

As part of the work in RAINS, a set of software tools was created to convert NXPol data into the Met Office NIMROD format using a combination of gridding software (Py-ART or LROSE) and bespoke scripts developed by NCAS. Many UK agencies (i.e. the Environment Agency and SEPA) use this format in their modelling and analysis tools such as HyRAD, developed by the UK's Centre for Ecology and Hydrology (CEH). These scripts may be requested from the authors.



97  **Figure 7. Observations and derived rainfall rates from the RAINS campaign on 20 July 2016 at 0409 UTC at 1.5° elevation: NXPol**

98  **is located at the black dot and range rings are drawn every 25km. (a) calibrated, corrected and filtered $Z_H$ classified as**

99  **precipitation echoes; (b), (c) and (d) calibrated, corrected and filtered $Z_{DR}$, $K_{DP}$, and $A_H$, (e) rainfall rate calculated using the**

00  **Marshall-Palmer relation (R(Z)=a$Z^b$, with a=200 and b=1.6) and (f) rainfall rates calculated using R($Z_H$,$K_{DP}$). NXPol is located at**

01  **the black dot and range rings are drawn every 25km.**

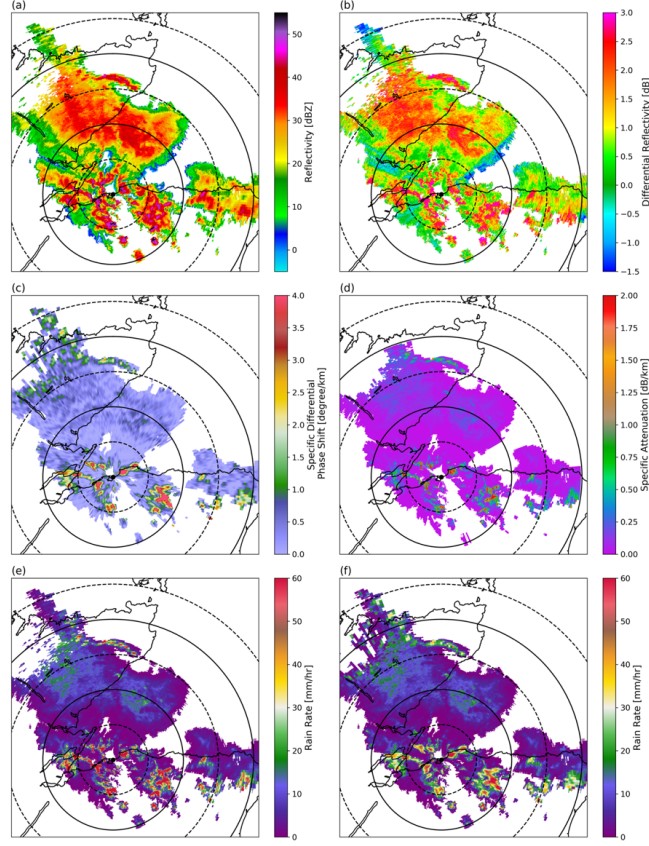

02




## 4 Ongoing Work at CFARR

In between major field deployments, the NXPol makes continuous observations at the Chilbolton Facility for Atmospheric Radar Research (CFARR), located near Andover in Hampshire. This enables NXPol to work in coordination with the other state-of-the-art radar facilities located at the observatory to make novel observations of high impact wintertime storms and summertime convective events using an array of ground-based remote sensing and in situ observations. The goal of this work is to improve flood forecasting in the UK by using these novel observations to drive the development of physical parameterisations in high-resolution numerical weather prediction models.

Most significantly, this work includes NXPol making coincidental RHI scans of frontal events with the Chilbolton Advanced Meteorological Radar (CAMRa), which is the largest steerable meteorological radar in the world. CAMRa operates at S-band (~3 GHz), and its 25m antenna creates a beam width of only 0.25°. This results in the ability to make high-resolution observations at far ranges (i.e. at 100 km from the dish, the resolution of a 0.25° beam is 0.4 km). Like the NXPol, CAMRa has dual-polarisation and Doppler capabilities. For a full description of CAMRa, please see Goddard et al. (1984). An example of coincidental observations from CAMRa and the NXPol on January 12$^{th}$, 2017 at 13:36 UTC are shown in Figure 8.

As part of the ongoing research with NXPol, the use of hydrometeor classification algorithms (HCAs; also referred to as particle identification or PID) to explore cloud microphysics is being pursued. Such an HCA has been initially implemented for the NXPol using the framework provided by LROSE (Dixon et al. 2012). The HCA is a fuzzy logic approach, and the membership functions are based largely on the work of Dolan and Rutledge (2009) and Thompson et al. (2014). An example result of the HCA applied to NXPol and CAMRa observations from May 17$^{th}$, 2017 at 12:24 UTC is shown in Figure 9. The NXPol's HCA results are part of on-going research and have not yet been fully validated. As such, Figure 9 is shown only to demonstrate the type of on-going investigations enabled by the NXPol's observations. Nevertheless, the comparison shows good qualitative agreement between the algorithms applied to the two radars. Future work will include validation with in situ observations made with FAAM.





**Figure 8. Coincident RHIs of Z$_H$ (a and b) and Z$_{DR}$ (c and d) from the NXPol (a and c) and CAMRa (b and d) on 12 January 2017**
**at 13:36 UTC.**

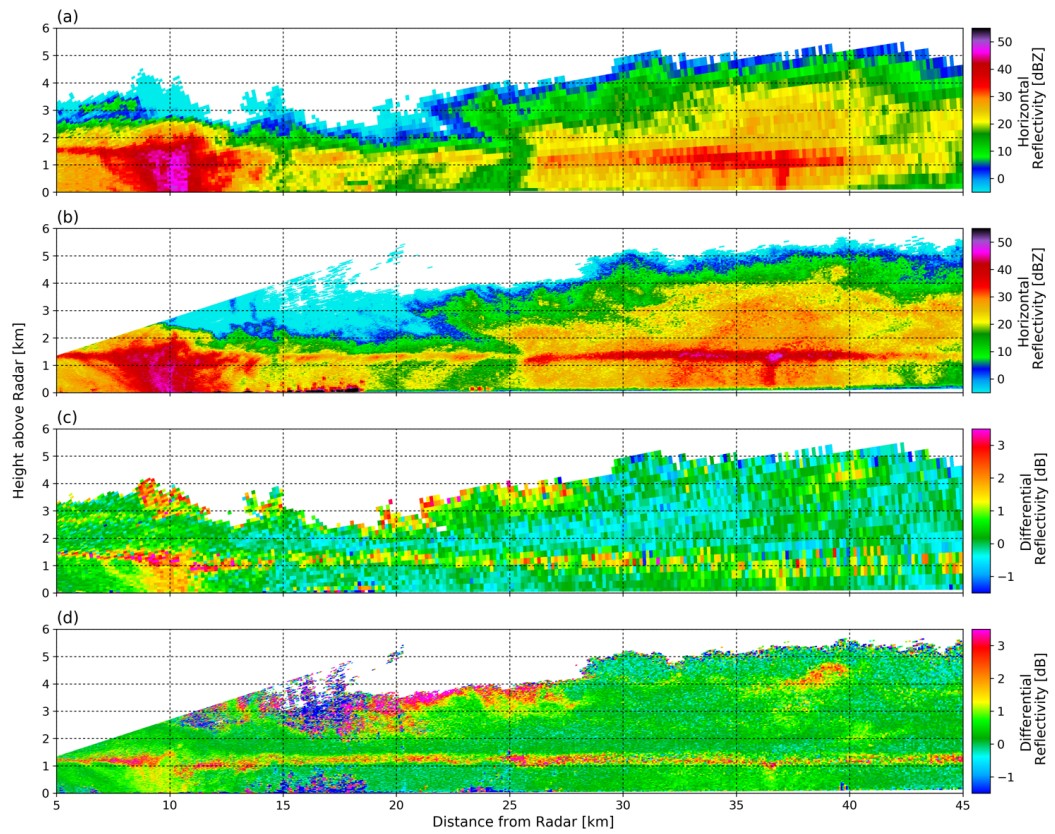




**Figure 9. Coincident RHIs from the NXPol (a) and CAMRa (b) on 17 May 2017 at 12:24 UTC with the HCA applied to both.**

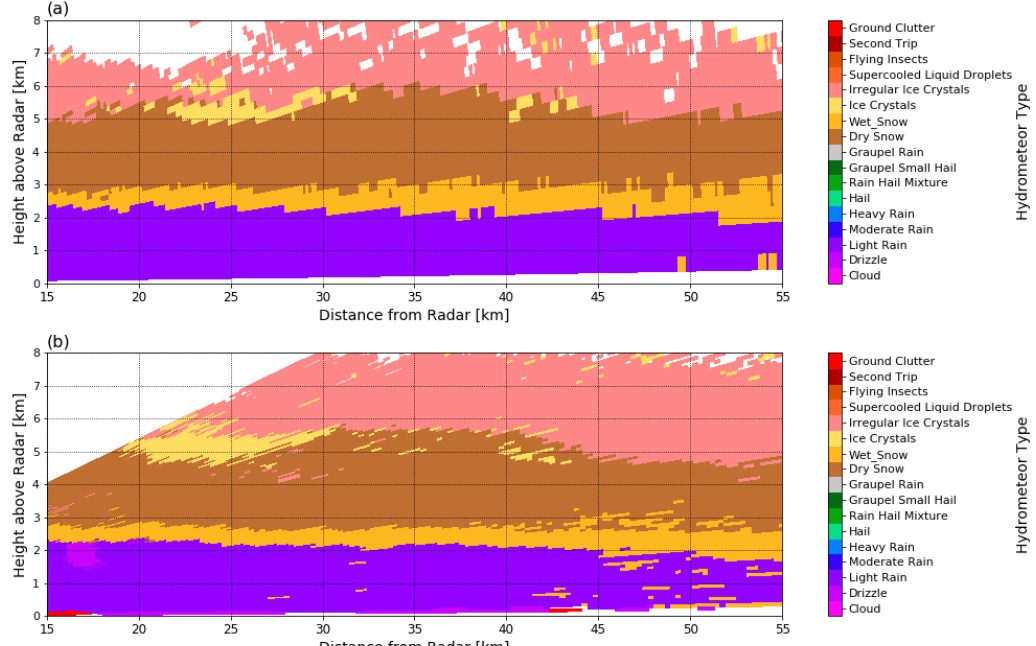




## 5 Summary

Here we have summarised the key technical characteristics of the NXPol and the infrastructure used to deploy the system autonomously at remote locations. We have also shown examples of its successful use in four differing scientific campaigns. As is shown in the examples, the NXPol is a highly capable and flexible instrument for use in examining the microphysics of clouds and producing QPE. As described in Section 4, in between bespoke deployments to remote locations, the NXPol will be located at CFARR to make continuous observations in conjunction with other instruments at this site. The NCAS and Leeds University Radar Group welcomes any collaborations that utilise the NXPol and its observations.

For further information on the use of the NXPol including instrument access policies, data format, NXPol specific analysis software and availability, please see the NXPol instrument homepage at: https://www.ncas.ac.uk/index.php/en/about-amf/263-amf-main-category/amf-x-band-radar/1098-x-band-radar-overview.

**Acknowledgements.**

Numerous people have provided assistance in the development and deployment of the NXPol since its purchase in 2012. We thank them for their contributions and support in this effort. In particular, we thank Selex ES Gmbh for their excellent support and Mike Dixon of NCAR's Earth Observing Laboratory for his continued help in adapting LROSE to the needs of the NXPol. The lead author would like to acknowledge the "Shut Up and Write" group in the School of Earth and Environment at the University of Leeds. Without the weekly space, time, and support this group offers, this manuscript would not have been written.



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
