# Peer review of "The NCAS Mobile Dual-Polarisation Doppler X-Band Weather"

_Atmospheric Measurement Techniques, 2018_

## Referee Comment (RC1) · Anonymous Referee #1 · 24 Apr 2018

General comments

The paper is well written and while the use of mobile x-band radars in this field is well known, this paper does a good job of highlighting the availability of this specific resource, and its use in several field campaigns. I believe the utility of the paper to the community would be enhanced by addressing the concerns described below.

Specific comments

Abstract.

13: The use of "prevalent" seems to overstate the use of X-band radars, particularly in relation to the QPE. Perhaps clarify that this is in the mobile / research campaign area.

[Figure]

AMTD

1. Introduction

27: Again "ubiquitous" seems to overstate the use of x-band, particularly in relation to the QPE. It could perhaps be stated that this is the case for mobile applications but that this is not clear from this statement.

2. Technical 54 Summary of the NXPol

2.1 Operations

75: Table 1 – Is the power per channel or pre-split? Please include the radar sensitivity. Ideally both of the receiver and the radar system as a whole.

03: DOP has been calculated for other systems previously (as per the work of Galletti, Bebbington, Holt, etc.) Please clarify how the calculation of DOP in this case is "unique". Also, this capability seems interesting but is not mentioned further; for example, is it used in the field campaigns described? Is it found to be a useful parameter?

26: "They were also invaluable" – While it is clear how the data could be used in the aircraft case, it would be useful to describe how the data were used by forecasters - in what way were they invaluable?

2.2 Deployment Setup

41: The increased ease of health and safety could be mentioned at this point – consider a forward reference to section 2.3.

65: "now" - when exactly is this?

2.3 Safety

02: It would be useful for others considering such a setup if some details of the contingency plan could be given – how is this issue addressed in practice?

3. Example Deployments and Observations

3.1 COPE

Figure 5: c) Spokes can be seen in the figure but are not referred to – what is this source of this artefact in this parameter alone?

Figure 5: d) the expanded colour bar label is difficult, if not impossible to read. Please revise.

3.2 ICE-D

Figure 6: Figure appears to be reversed. Presumably the thick black lines represent geographical features (islands) rather than meteorological ones – not actually stated. Is this data set publicly available? If so where? – If not please clarify the point being made in this and subsequent sections.

3.3 Radar Applications in Northern Scotland (RAINS)

88: It is unclear what conclusion one is to draw from the QPE in this figure; other than that different algorithms give different results - can this be clarified? It would be useful to state that one is making use of Kdp in the text rather than this having to be picked up from the figure label. Is a particular algorithm being used with the Kdp case? Does the use of Kdp improve comparison against ground truth?

Again – is this data set available? Were any conclusions drawn regarding the benefits of the X-band data in this area?

4. Ongoing Work at CFARR

Is this high quality dual wavelength data set available and if so, from where?

It is unclear to the reader, what the benefit/use of the lower resolution, more attenuation prone X-band data is in this case. i.e. what is gained by this dual wavelength validation of the HCA?

---

## Referee Comment (RC2) · S. Collis (Referee) · 14 May 2018

This is one of those "necessary"papers to document a community resource so that others can cite and reference when using the system.

The paper is well written and despite the documentary nature it reads well. The main issue I would like to see addressed is better documentation of the sensitivity.

At the start of section two phenomena detectable by the radar are discussed but no examples of minimum detectable signal (MDS) are given.

Please add (at a minimum) information about the pre-integration (single pulse) MDS at 10km (for example) to table 2 for the same waveform configuration.

In addition please modify line 58 as insects are not clear air returns. This can mislead some audiences to thinking you can see Bragg returns with the radar. Use words like non-meteorological or Aeroplankton.

My last comment is a suggestion and not a requirement: The colormaps used in your manuscript are very unfriendly to scientists with Color Vision Deficiency (CVD). I would recommend using perceptually uniform colormaps. I fully concede this is not common practice within the radar community 8% of the male population have CVD including (from my study) many well known radar meteorologists. For an interesting discussion on the introduction of a CVD colormap to Py-ART see: https://github.com/ARM-DOE/pyart/issues/713

---

## Author Comment (AC1) · 10 Jul 2018

**Response to RC1**

The authors thank the reviewer for the time and consideration given this manuscript. We appreciate your overall view that the paper will benefit the community and are ahppy that the description

The reviewers comments have been listed below in **bold** and responded to individually (where needed) in *italics*.

**General comments**
**The paper is well written and while the use of mobile x-band radars in this field is well known, this paper does a good job of highlighting the availability of this specific resource, and its use in several field campaigns. I believe the utility of the paper to the community would be enhanced by addressing the concerns described below.**

**Specific comments**
**Abstract.**
**13: The use of "prevalent" seems to overstate the use of X-band radars, particularly in relation to the QPE. Perhaps clarify that this is in the mobile / research campaign area.**
*Response: We agree with the reviewer and have replaced "prevalent" with "widely" in the abstract in addition to adding a follow-on sentience for clarification about the primary use in mobile applications. The revised section of the abstract is as follows:*
*"Abstract. In recent years, dual-polarisation Doppler X-band radars have become a widely used part of the atmospheric scientist's toolkit for examining cloud dynamics and microphysics and making quantitative precipitation estimates. This is especially true for research questions that require mobile radars."*

**1. Introduction**
**27: Again "ubiquitous" seems to overstate the use of x-band, particularly in relation to the QPE. It could perhaps be stated that this is the case for mobile applications but that this is not clear from this statement.**
*Response: As with the comment above, we agree with the reviewer and have made the clarification with the abstract as follows:*
*"Thus, small and or mobile dual-polarisation Doppler X-band radars have become popular tools for examining cloud microphysics and dynamics as well as making quantitative precipitation estimates (QPE) in mobile applications (Wurman et al. 1997; Matrosov et al. 2005; Wang and Chandrasekar, 2010)."*

**2. Technical 54 Summary of the NXPol 2.1 Operations**
**75: Table 1 – Is the power per channel or pre-split? Please include the radar sensitivity. Ideally both of the receiver and the radar system as a whole.**
*Response: We have now clarified in the table that this is pre-split and have added the information about the sensitivity of the radar to Table 1.*

**103: DOP has been calculated for other systems previously (as per the work of Gal- letti, Bebbington, Holt, etc.) Please clarify how the calculation of DOP in this case is "unique".**

*Response: We agree with the reviewer that the use of the word "unique" is over stated in this instance given the previous work. It was intended to mean that it was unique in comparison to other Meteor 50/60DX radars at the time it was purchased. The section of text has been revised as follows:*
*"In addition to the standard polarimetric variables provided by most operational dual-polarisation radars, NXPol also provides the of the degree of polarisation (DOP) of the backscattered signal."*

**Also, this capability seems interesting but is not mentioned further; for example, is it used in the field campaigns described? Is it found to be a useful parameter?**
*Response: DOP has been collected in the campaigns described within the paper but has only begun to be explored for its use in HCA due to its advantages over co-polar correlation coefficient for STAR mode radars. Currently we are not in the position to answer the question about its usefulness further or provide any further information. It is included here only to fully document the capability of the NXPol.*

**26: "They were also invaluable" – While it is clear how the data could be used in the aircraft case, it would be useful to describe how the data were used by forecasters - in what way were they invaluable?**
*Response: We agree with the reviewer that no reason for the vlaue of the quick-look images was originally given and have clarified the sentence as below:*
*"The quick-look images were also helpful to the NXPol's operators and forecasters at the Scottish Environmental Protection Agency and the U.K.'s Met Office during the six-month-long Radar Applications in Northern Scotland (RAINS) campaign in 2016 (Section 3.3) for assessing the impact that a radar in this location would have on observational quality in near-real-time conditions in comparison to existing observations."*

**2.2 Deployment Setup**
**41: The increased ease of health and safety could be mentioned at this point – consider a forward reference to section 2.3.**
*Response: We agree, and this has now been included.*

**65: "now" - when exactly is this?**
*Response: We agree with the reviewer that this usage of "now" makes no sense and have revised the sentence to the following:*
*"The RAINS project and on-going work at NFARR (where the NXPol will operate for several months at a time between campaign deployments) in particular demonstrate the need for this type of facility to support the radar in its long-term operations. "*

**2.3 Safety**
**02: It would be useful for others considering such a setup if some details of the contin-gency plan could be given – how is this issue addressed in practice?**
*Response: We feel this sentence was ambiguous and has now been removed and replaced with the following information:*
*"If the radar is unmanned, then access must be restricted to the distance at which public exposure limits (10W/m$^2$) are reached in the event the NXPol malfunctions and stops scanning but continues to transmit."*

**3. Example Deployments and Observations 3.1 COPE**
**Figure 5: c) Spokes can be seen in the figure but are not referred to – what is this source of this artefact in this parameter alone?**
*Response: The spokes are due a quality control filter on the calculation of Ah. This has been noted in the text.*

**Figure 5: d) the expanded colour bar label is difficult, if not impossible to read. Please revise.**
*Response: We have doubled the original size of the colour bar label text.*

**3.2 ICE-D**
**Figure 6: Figure appears to be reversed. Presumably the thick black lines represent geographical features (islands) rather than meteorological ones – not actually stated.**
*Response: This is now stated in the caption of the figure.*

**Is this data set publicly available? If so where? – If not please clarify the point being made in this and subsequent sections.**
*Response: Yes, this data is available on CEDA. Its location has now been included in the text.*

**3.3 Radar Applications in Northern Scotland (RAINS)**
**88: It is unclear what conclusion one is to draw from the QPE in this figure; other than that different algorithms give different results - can this be clarified?**
*Response: As the point of this paper is to show what types of research the NXPol can be used for, that is exactly the conclusion we want people to draw. Further results will be given in a publication focused on the results of the SEPA campaign.*

**It would be useful to state that one is making use of Kdp in the text rather than this having to be picked up from the figure label.**
*Response: We Agree and this has not been included in the main text.*

**Is a particular algorithm being used with the Kdp case?**
*Response: The KDP relation used is that of Diederich et al. 2015 using a = 16.9 and  b = 0.801. This is stated in the text.*

**Does the use of Kdp improve comparison against ground truth?**
*Response: The answer to this question is part of on-going research and is beyond the scope of this paper. We intend this section to only depict an example of how the radar could be used to look at questions related to QPE rather than answering those questions.*

**Again – is this data set available?**
*Response: Currently, this dataset may be requested from the author as it is still undergoing its primary analysis with SEPA.  The dataset will become public on CEDA in the coming year when the first of 2 papers have been submitted.  This has been clarified in the text.*

**Were any conclusions drawn regarding the benefits of the X-band data in this area?**

*Response: As the response to the comment above states, the goal of this manuscript is just to provide examples of how NXPol is used. It is not to describe results of specific studies. In addition, we are in the midst of finalising the analysis of the RAINS project but due to limits placed on us by SEPA, we are unable to say any more.*

**4. Ongoing Work at CFARR**
**Is this high quality dual wavelength data set available and if so, from where?**
*Response: Currently, this dataset may be requested from the author as it is still undergoing its primary analysis. The dataset will become public on CEDA in the next year. This has been clarified in the text.*

**It is unclear to the reader, what the benefit/use of the lower resolution, more attenuation prone X-band data is in this case. i.e. what is gained by this dual wavelength validation of the HCA?**
*Response:* The benefits of dual-wavelength validation of the HCA are the constraints the multiple frequencies place on the scattering parameters. This has been clarified in the text.

[revised manuscript text omitted]

---

## Author Comment (AC2) · 15 Jul 2018

**Response to RC2**

The authors thank the reviewer for the time and consideration given this manuscript. We appreciate your overall view that the paper will benefit the community and are appry that the description

The reviewers comments have been listed below in **bold** and responded to individually (where needed) in *italics*.

This is one of those "necessary" papers to document a community resource so that others can cite and reference when using the system.

**The paper is well written and despite the documentary nature it reads well.** *Response: We very much appreciate this feedback from the reviewer.*

The main issue I would like to see addressed is better documentation of the sensitivity. At the start of section two phenomena detectable by the radar are discussed but no examples of minimum detectable signal (MDS) are given.

Please add (at a minimum) information about the pre-integration (single pulse) MDS at 10km (for example) to table 2 for the same waveform configuration.

Response: As was requested by RC1 we have added this information.

In addition please modify line 58 as insects are not clear air returns. This can mislead some audiences to thinking you can see Bragg returns with the radar. Use words like non-meteorological or Aeroplankton.

*Response: We agree with the reviewer and have changed the use of "clear air" to nonmeteorological in all instances.*

My last comment is a suggestion and not a requirement: The colormaps used in your manuscript are very unfriendly to scientists with Color Vision Deficiency (CVD). I would recommend using perceptually uniform colormaps. I fully concede this is not common practice within the radar community 8% of the male population have CVD including (from my study) many well known radar meteorologists. For an interesting discussion on the introduction of a CVD colormap to Py-ART see: https://github.com/ARM-DOE/pyart/issues/713

Response: We have attempted to comply with this suggestion to the best of our ability in all figures. In particular we have switch from using 'pyart\_NWSRef' to the newly created 'pyart\_HomeyerRainbow' mentioned in the threads referenced above.

**The NCAS Mobile Dual-Polarisation Doppler X-Band Weather 1**

**Radar (NXPol)** 2**

Ryan R. Neely III1,2, Lindsay Bennett1,2, Alan Blyth1,2, Chris Collier1,2, David Dufton1,2, James 3 Groves1,2, Daniel Walker1,2, Chris Walden1,3,4, John Bradford1,3,4, Barbara Brooks1,2, Freya Lumb1,2, 4 John Nicol1, Ben Pickering1,2 5 6 1National 
[revised manuscript text omitted]

:09

**11**

**Formatted Table**